# Flatfoot and associated factors among Ethiopian school children aged 11 to 15 years: A school-based study

Yohannes Abich[1], Tewodros Mihiret[1], Temesgen Yihunie Akalu[2], Moges Gashaw[1], Balamurugan Janakiraman[1]*

1 Department of Physiotherapy, College of Medicine and Health Sciences & Comprehensive Specialized Hospital, University of Gondar, Gondar, Ethiopia, 2 Department of Epidemiology and Biostatistics, Institute of Public Health, College of Medicine and Health Sciences, University of Gondar, Gondar, Ethiopia

* bala77physio@gmail.com

**Data Availability Statement:** All data relevant to our findings are contained within the manuscript. The full datasets contain sensitive participant information and ethical restrictions. Requests for

## Abstract

### Background

The Foot health of a child plays a pivotal role in their participation in play, locomotive activities, healthy lifestyle, somatic development, and weight management. The burden of flatfoot among children in Ethiopia is not known. The objective of this study was to analyze the structure of the medial foot arch using Staheli plantar arch index and investigate its associated factors among larger sample school children, aged 11–15 years in Ethiopia.

### Methods

A school-based cross-sectional study was conducted among children aged 11–15 years from eleven randomly selected primary schools. The sample size was determined proportionally across school strength and governmental and private schools to ensure variety within the sample. Data collection consisted of physical measurements, footprint-based measures whilst full weight-bearing, and a structured questionnaire on foot pain, footwear type, and physical activity. Data were analyzed descriptively and through uni- and multivariate logistic regression model.

### Results

A total of 823 children participated. The overall prevalence of flatfoot was 17.6% with a significant difference between age, gender, type of school, BMI, and type of footwear. Being younger (OR 3.3, 95% CI 1.6–6.7), male (OR 1.6, 95% CI 1.0–2.4), experiencing foot pain (OR 1.9, 95% CI 1.0–3.5), wearing closed shoe (OR 4.4, 95% CI 1.6–11.9), overweight (OR 3.8, 95% CI 1.2–8.7), obese (OR 4.2, 95% CI 2.5–10.9), and low level of physical activity (OR 2.1, 95% CI 1.0–4.6) were significantly associated with flatfoot. Children who were overweight, obese, and also experiencing foot pain have a 2.8 (95% CI 1.62–5.94) and 4.1 (95% CI 2.85–8.31) times greater chance of having flatfoot respectively. The prevalence of flatfoot among 560 normal weight children was 17.5%.

further details on the datasets and queries concerning data sharing shall be arranged based on a reasonable request to the corresponding author bala77physio@gmail.com.

**Funding:** This study was fully funded by the University of Gondar (Grant no: SOM 1251/2019). The funders had no role in study design, data collection and analysis, decision to publish, or preparation of the manuscript.

**Competing interests:** The authors have declared that no competing interests exist.

**Abbreviations:** BMI, Body Mass Index; CDC, Centre for Disease Control; EDHS, Ethiopian Demographic and Health Survey; GPI, ender Parity Index; LMICs, Low-Middle Income Countries; SPAI, Staheli Plantar Arch Index.

## Conclusions

The findings of this study demonstrated that the overall prevalence and the prevalence of flatfoot among normal-weight children are almost the same. There is a definite need to develop a screening algorithm for diagnosis and treatment indication for this children's lower extremity disorder.

## Background

A healthy foot is vital for good posture and ambulation. Flatfoot or pes planus is a medical condition defined by the absence or lowered medial longitudinal arch, with osseo-ligamentous misalignment [1]. Due to plantar fat pad, the infant's feet appear flat; this fatty pad disappears between 2 and 10 years of age after developmental changes in the medial arch [2]. The anatomical manifestations of flattening or lowering of medial foot-arch are ligamentous laxity, equinus deformity, torsional deformity, vertical talus, and tarsal coalition, caused by multifactorial variables like overweight, obesity, type of footwear's, weak muscles that support the arch, foot injury, and congenital deformations [3–6]. Besides, many authors claim that flatter foot structure among obese school-aged children might be due to fat feet rather than the structural lowering of the arch (flat feet) and emphasized the need to include a larger sample of normal weight control participants to explain the association in the absence of imaging outcome measures [7–9]. Flatfoot often leads to tissue strain, pain, acceleration of overuse injuries, poor functions, and disability in lower extremities which has a negative impact on the quality of life of children [10]. In particular, attention should be given to the rising prevalence of childhood overweight and obesity in Ethiopia [11], as these populations tend to have a higher risk of flatfoot.

In the presence of flatfoot, there is a biomechanical alteration of the center of gravity and kinetic chain of the body, which increases stress on the joint structures of the spine, hip, knee, and ankle causing gait and postural defects in all age groups [12]. The growing structures in children are more vulnerable to these changes, flatfoot among children is also a major parental concern, and one of the most frequently reported reasons to seek a pediatric specialist or orthopedic opinion hence, shall not be ignored. About 90% of all clinical visits related to foot problems in children are due to flatfeet [13, 14].

The reported global prevalence of flatfoot among children varies based on countries, age groups, diverse evaluation methods, and nutritional status of the children population, which widely ranges from as low as 0.6 to as high as 77% [2]. In Sub-Saharan Africa, the reported prevalence of childhood flatfoot in Kenya, Nigeria, and Tanzania is 45.3%, 22.4%, %, and 20.3% respectively [15, 16]. Besides, a regional study reported that 7.8% of schoolchildren had ankle and foot pain [17]. Studies had reported several risk factors for flatfoot among children like family history, age, gender, weight, BMI, type of footwear, physical activity, and associated with hypermobility, genu valgum, and heel valgus [1, 5, 6, 12, 18]. The critical time for the development of the plantar arch is before 6 years, continues to develop until the age of 10and normative data indicates 'flat' is normal for children up to 10 years of age [2, 12].

Hence, we thought, a sample of age 11–15 comprising a large proportion of normal weight school children, and using the standardized, preferred method of foot posture measurement (Staheli arch index) [2, 19] will surely provide us with close to accurate estimation of the prevalence. Moreover, already prevalent congenital deformities and the rising prevalence of childhood overweight, obesity, in Ethiopia, and the deteriorating effects of overload on the growing

foot arch is well known. The lack of data on the burden of flatfoot among Ethiopian children is concerning and demands greater attention "every step on the deformed foot leads to further irreversible damage". We sturdily believe that the methods used and results of this study will help the Ethiopian health professional in identifying when or not a child's foot is developing normally, factors associated, monitor, or even intervene if needed. Therefore, this study aimed to determine the prevalence of flatfoot using the Staheli plantar arch index (SPAI) of footprints and identify factors associated with Ethiopian school children aged 11 to 15 years.

## Material and methods

### Study design, setting, and population

A school based cross-sectional study design was conducted among public and private primary-school-age children between 11–15 years in Gondar town, northwest Ethiopia. Gondar town is located at a latitude and longitude of 12º36'N and 37º28'E respectively, with an elevation of 2133 meters above sea level in the northern part of the Amhara regional state, northwest Ethiopia about 727 km from Addis Ababa, the capital city of Ethiopia. According to the education authority office of the Gondar town, 25,880 school children were attending 34 public and 17 private primary schools in Gondar city in the 2018/19 academic year. According to the Ethiopian Demographic and Health Survey (EDHS, 2016), the primary school net attendance ratio (NAR) is 71% (72% girls and 71% for boys). However, in the Amhara region, this national report revealed a slightly higher NAR 75.6 for primary school and a better gender parity index (GPI) than the overall country's GPI, 1.08 of the Amhara versus 1.01 of Ethiopia [20].

The population of interest comprised of all school-age children between 11–15 years of both sexes in Gondar town. School children with parental consent, children assent, and able to ambulate independently were included. Children with known or evident lower disorders, lower limb or foot deformity, lower limb weakness or paralysis, recent lower injuries, and recent lower extremity surgeries that would hamper the accurate foot outcome were excluded.

### Sample size and sampling technique

In total 25,880 elementary school children were registered as a student in private and government schools in Gondar city by the local government school authority bureau. So, the sample was taken from this registered population (N = 25,880). The following assumptions were used to determine the sample size based on a single population proportion; 50% proportion of flatfoot since no past regional data exist, the confidence level of 95%, and a design effect of 2. Accounting for an estimated non-response or contingency or refusal of 10%, the derived sample size was n = 845.

A multistage sampling procedure was applied. The schools were stratified into governmental and private elementary schools. After that, 11 schools (7 governmental and 4 private) were proportionally selected lottery method. Based on the student population the participants were proportionally selected from each selected school using a systematic sampling method for which the name register was used. The $K^{th}$ was calculated by dividing the total number of students in each section by the number of school children included in the study. The first participant was selected randomly between and $K^{th}$, the next children were chosen with K value until the required sample was achieved (Fig 1). After the children were identified with flatfoot, parents were contacted through the school authorities and referred to visit the department of physiotherapy at the University of Gondar teaching hospital.

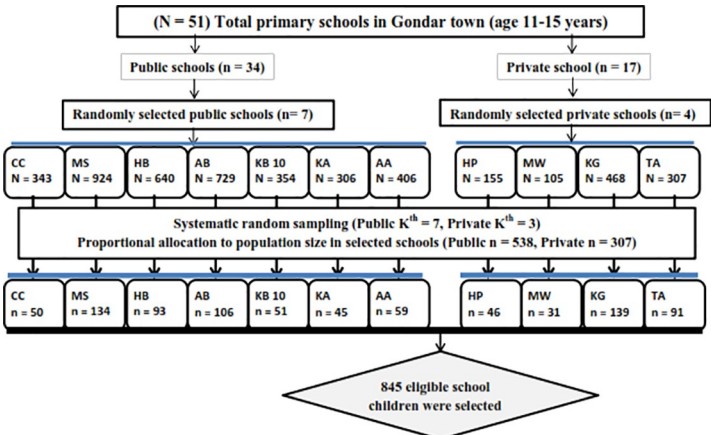

**Fig 1. Schematic presentation of sampling procedure for flatfoot and associated factor among school children in Ethiopia.**

## Data collection tools and procedures

A structured questionnaire was used to record socio-demographic data such as age, gender, and type of school. Measurement of children's weight (kg) was done using a digital weighing scale (Electrolux, Korea) to the nearest 1 kg without shoes, and height was measured to the nearest 1 cm using a stadiometer (S1 File). Weight was first measured with children standing on the weighing scale wearing their school bag and then again without their bag. The difference between the two recordings was recorded as school bag weight. Then the feet were thoroughly cleaned and dried, with the children in a sitting position was asked to dip the foot into the ink tray. The participant was then asked to remove the foot from the tray and stand up to print the foot firmly on the sheet on the wooden platform with about 50% weight-bearing, the procedure was repeated to print the other foot.

The SPAI was then calculated using the footprint of the children for each foot, after labeling the sheet with participant data unique id number and foot side (right and left). Using a pencil a line was drawn tangential to the medial border of the forefoot and hindfoot; the midpoint of these two lines was marked. Then the narrowest printed part of midfoot was identified both visually and using a ruler, the perpendicular distance (line A) representing the width the narrowest part of midfoot is marked. The second perpendicular line (line B) was drawn representing the width of the hindfoot, the measurement of line A and line B is noted (Fig 2).

The SPAI was then calculated by the Staheli arch index method [21] by dividing the value of line A by the value of line B (AI = A/B) and the ratio between these widths is called Staheli's Arch Index (AI). Any participant with Staheli arch index >1.15 [22] was considered to have flatfoot. Participants who self-reported foot pain in the past 6-months and diagnosed by FPI as flatfoot are defined as symptomatic flatfoot, those diagnosed with flatfoot without pain are defined as asymptomatic flatfoot. Participants diagnosed with flatfoot by footprint were asked to perform tip-toe standing and using observation method the children with flatfoot were classified as flexible or rigid flatfoot based on the appearance of arch.

Prior to data collection, two-day intensive training was given to four data collectors on the data collection procedure by the principal investigator. The data collection process was supervised by the primary investigator on a daily basis to ensure its completeness, accuracy, and clarity. Ethical clearance was obtained from the ethical review committee of the School of

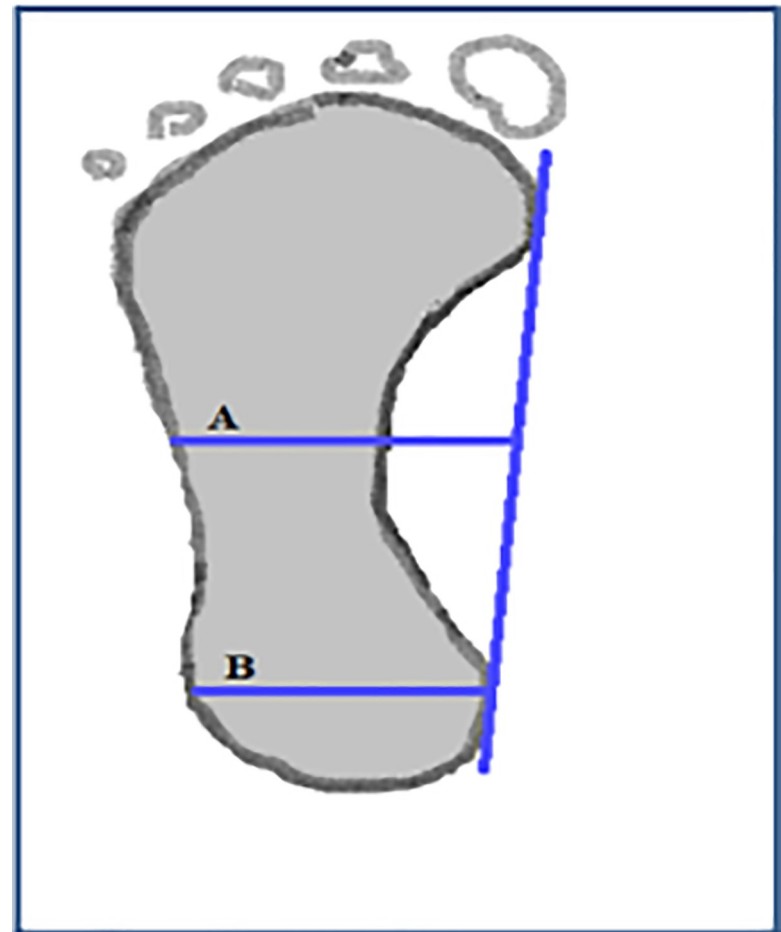

**Fig 2. Schematic illustration of Staheli plantar arch index calculation.** 'A' represents the width the narrowest part of midfoot, 'B' represents the width of the hindfoot.

Medicine, College of Medicine and Health Science, University of Gondar (SOM/1251/2019). An official letter was sent to the respective body in the selected schools requesting permission to conduct the study. All the study participants and parents were oriented on the objectives and purpose of the study prior to study participation. Informed consent was obtained from parents and children. The participant's involvement was voluntary. Confidentiality and anonymity were explained.

## Data analysis

The collected data were coded, re-checked for completeness, and entered into Epi Info software version 7.0, and then exported to IBM Statistical Package for Social Sciences (SPSS) version 20 windows for statistical analyses. Descriptive statistics (frequencies, percentages, means, and standard deviations) were expressed for all the participant characteristics and associated factors of flatfoot. Body Mass Index (BMI) was categorized as: underweight ($< 5^{th}$ percentile), normal weight ($5^{th}$ percentile to $< 85^{th}$ percentile), overweight ($85^{th}$ to $< 95^{th}$ percentile), and obese ($\geq 95^{th}$ percentile) [23] based on age and sex-specific Centre for Disease Control (CDC) growth chart for children. Bivariate logistic regression analysis was done to identify factors associated with flatfoot (>1.15 Staheli arch index as flatfoot present versus $\leq 1.15$ as none)

independently at a P-value < 0.2. Variables that were significant at a p-value of < 0.2 were selected for the final multivariate logistic regression model to control the possible confounds and to examine the association between different independent variables. In multivariate analysis variables with 95% confidence intervals and a p-value of < 0.05 were considered statistically significant. Potential confounding variables were entered into the model as covariates and variability in the association was examined. When clear subgroups per category existed in the data, significance testing (Pearson $\chi^2$) and logistic regression model were performed for appropriately sized subgroups. Interaction terms were used to examine the potential association between BMI, weight, and flatfoot differed by hypothesized variables, including age, type of shoe, physical activity, and foot pain. Finally, the research was checked for adherence to STROBE guidelines (S2 File).

## Results

### Sample characteristics and prevalence

A total of 845 school children were approached for participation, out of which 823 (351 male, 472 female) parents consented for their children to participate, with a total of 1646 feet screened. The response rate was 97.4% and this is 100% more than the power calculated sample size (n = 768). The common reason for non-response was the absence of parental consent form followed by not willing for ink footprint. The mean age (in years), height (cm), weight (kg), and BMI (kg/m$^2$) of the participants were 13.2, 139.6, 39.7, and 20.2 respectively. As of, CDC age–gender-adjusted BMI, most of the children were in normal weight (68%) followed by underweight (15.6%), overweight (11.3%), and obese (5.1%).

The majority of children (82.4%) had normal plantar arch while the overall prevalence of flatfoot was (n = 145) 17.6%, of which about 80% were flexible flatfoot, and with increasing age, the prevalence of flatfoot decreased. The most common footwear type worn by the participants (67.9%) were sandals and only 10.6% of the children reported to be involved in high-level physical activity. About 1/4$^{th}$ of the participants self-reported to have experienced foot pain in the past 6 months and one in three children with foot pain had a flat foot. More sample characteristics are presented in Table 1.

### Associated factors of flatfoot

Self-reported foot pain between the categories of BMI was vastly different ($\chi^2$ (3, n = 823) = 156, p<0.001, phi = 0.19), with most of the obese children reporting foot pain (66.7%) followed by overweight children (62.1%), and nearly 70% of normal and underweight children diagnosed with normal foot reported no foot pain (n = 101, 69.3% versus 30.7%). The proportion of underweight children was highest in governmental schools 17.5% (n = 93) and obese children in private schools 6.2% (n = 18), the BMI distributions significantly varied between government and private schools ($\chi^2$ (3, n = 823) 532 versus 291, p<0.001, phi = 0.17). Similarly, the trend of symptomatic flatfoot (flatfoot with foot pain) increased with increasing BMI among normal weight, overweight, and obese, 31.6%, 62.1%, and 66.7% respectively Table 2.

The prevalence of flatfoot was highest (30.1%) among children aged 11 years and lowest (9.9%) in 15 year age group. There was a significant association between age, gender, BMI, type of shoe worn, foot pain, physical activity level, and flatfoot. Sandals were the most common footwear worn (n = 559, 67.9%) by the children, with most children from government schools wearing sandals (n = 419, 78.8% versus 21.2%). Of those in private school, 51.9% (n = 151) worn closed-toe shoes, which constitute a subgroup in which the prevalence of flatfoot was found (35.1%, n = 53). Among the 19.8% (n = 156) participants who self-reported

**Table 1. Socio-demographic, physical measurements, behavioral characteristics, and distribution of flatfoot (Staheli arch index > 1.15) elementary school children in Gondar town, Ethiopia (n = 823).**

| Variables | Sample total | Flatfoot n (%) | | $\chi^2$ | p |
|---|---|---|---|---|---|
| | n (%) | None | Present | | |
| **All participants** | 823 (100) | 678 (82.4) | 145 (17.6) | | |
| **Age in years** | | | | | |
| 11 | 113 (13.7) | 79 (69.9) | 34 (30.1) | 25.3 | .000* |
| 12 | 172 (20.9) | 135 (78.5) | 37 (21.5) | | |
| 13 | 169 (20.5) | 136 (80.5) | 33 (19.5) | | |
| 14 | 198 (24.1) | 174 (87.9) | 24 (12.1) | | |
| 15 | 171 (20.8) | 154 (90.1) | 17 (9.9) | | |
| **Gender** | | | | | |
| Male | 351 (42.6) | 274 (78.1) | 77 (21.9) | 7.86 | .005* |
| Female | 472 (57.4) | 404 (85.6) | 68 (14.4) | | |
| **Types of school** | | | | | |
| Government | 532 (64.6) | 454 (85.3) | 78 (14.7) | 9.06 | .003* |
| Private | 291 (35.4) | 224 (77) | 67 (23) | | |
| **BMI** | | | | | |
| Underweight | 128 (15.6) | 125 (97.7) | 3 (2.3) | 41.8 | .000* |
| Normal | 560 (68) | 462 (82.5) | 98 (17.5) | | |
| Over weight | 93 (11.3) | 64 (68.8) | 29 (31.2) | | |
| Obese | 42 (5.1) | 27 (64.3) | 15 (35.7) | | |
| **Type of shoe** | | | | | |
| Close toe shoe | 264 (32.1) | 177 (67) | 87 (33) | 62.9 | .000* |
| Sandals | 559 (67.9) | 501 (89.6) | 58 (10.4) | | |
| **Wearing shoes before 6 years** | | | | | |
| Yes | 748 (90.9) | 615 (82.2) | 133 (17.8) | .149 | .70 |
| No | 75 (9.1) | 63 (84) | 12 (16) | | |
| **Physical activity level in min/wk** | | | | | |
| Low (<180 min/week) | 736 (89.4) | 602 (81.8) | 134 (18.2) | 1.65 | 0.019 |
| High (>180 min/week) | 87 (10.6) | 76 (87.4) | 11 (12.6) | | |
| **Foot pain** | | | | | |
| Yes | 156 (19.8) | 97 (62.2) | 59 (37.8) | 54.1 | .000* |
| No | 667 (80.2) | 581 (87.1) | 86 (12.9) | | |

*Denotes statistical significance (p<0.05)

foot pain, majority of them reported having experienced foot pain rarely followed by occasionally. Most of them experienced foot pain while walking (n = 87), followed by playing and at rest.

**Table 2. Distribution of symptomatic (foot pain) and asymptomatic flatfoot based on BMI category among school children in Gondar town, Ethiopia.**

| BMI category | Flatfoot (n = 145) | | $\chi^2$ | P |
|---|---|---|---|---|
| | Asymptomatic (n = 86) | Symptomatic (n = 59) | | |
| Underweight | 3 (100) | 0 (0.0) | 15.08 | 0.002* |
| Normal weight | 67 (68.4) | 31 (31.6) | | |
| Overweight | 11 (37.9) | 18 (62.1) | | |
| Obese | 5 (33.3) | 10 (66.7) | | |

Among the children, the mean SPAI value for right and left foot was 0.74 and 0.68 respectively. The distribution of prevalence of flatfoot was; right foot (n = 58, 7%), left foot (n = 27, 3.3%), and bilateral (n = 60, 7.3%). There is a significant positive correlation between weight, BMI, and SPAI. Age and height showed a significant negative correlation with SPAI. Pearson's correlation for the association between age group, height, weight, BMI between right and left foot are shown in the S3 File.

## Regression analysis

Before conducting the regression model the following assumptions were tested: multi-collinearity (Variance inflation factor <10) independence of residuals by scatter plots. Initially, bivariate analysis was conducted with a cut-off point ($p < 0.25$), and independent variables that were found statistically significant were fitted into the multivariate model using the backward stepwise (likelihood ratio) method. Multivariable logistic regression (Table 3) was conducted by taking variables; age, gender, type of school, height, weight, BMI, foot pain, type of

**Table 3. Bivariate and multivariate analysis of factors associated with flat foot among school children in Gondar town, Ethiopia (n = 823).**

| Variables | COR (95% CI) | P | AOR (95% CI) | P |
|---|---|---|---|---|
| **Age** | | | | |
| 11 | 3.89 (2.05, 7.41) | 0.000 | 3.27 (1.60, 6.65) | 0.001** |
| 12 | 2.48 (1.33, 4.61) | 0.004 | 3.30 (1.65, 6.62) | 0.001** |
| 13 | 2.20 (1.17, 4.12) | 0.014 | 1.97 (1.01,3.97) | 0.048* |
| 14 | 1.25 (0.64, 2.41) | 0.507 | 1.12 (0.53,2.32) | 0.76 |
| 15 | 1 ref | | | |
| **Gender** | | | | |
| Male | 1.67 (1.16, 2.39) | 0.005 | 1.59 (1.03, 2.43) | 0.03* |
| Female | 1 ref | | 1 ref | |
| **School type** | | | | |
| Government | 1 ref | | 1 ref | |
| Private | 1.74 (1.21, 2.50) | 0.003 | 0.88 (0.55,1.39) | 0.59 |
| **Foot pain** | | | | |
| Yes | 4.11 (2.77, 6.10) | 0.000 | 1.9 (1.02,3.45) | 0.04* |
| No | 1 ref | | 1 ref | |
| **Type of foot wear*foot pain** | | | | |
| Sandals | 1 ref | | | |
| Closed shoe | 13.62 (7.78, 23.87) | 0.000 | 4.40 (1.62, 11.96) | .004** |
| **BMI** | | | | |
| Underweight | 1 ref | | | |
| Normal weight | 8.8 (2.75, 28.35) | 0.000 | 2.34 (1.21,7.44) | .001** |
| Overweight | 18.8 (5.54, 64.35) | 0.000 | 3.77 (1.23, 8.71) | .002** |
| Obese | 23.14 (6.26, 85.57) | 0.000 | 4.16 (2.51, 10.96) | .002** |
| **BMI*Foot pain** | | | | |
| Normal*foot pain | 3.59 (2.20, 5.85) | 0.031 | 1.31 (1.01, 4.57) | 0.06 |
| Overweight*foot pain | 9.62 (4.55, 20.34) | 0.020 | 2.85 (1.62, 5.94) | 0.03* |
| Obese*foot pain | 9.93 (3.68, 26.78) | 0.001 | 4.11 (2.85, 8.31) | 0.012* |
| **Type of footwear** | | | | |
| Sandals | 1 ref | | | |
| Closed shoe | 4.25 (2.92, 6.17) | 0.00 | 2.59 (1.57, 4.29) | 0.000** |
| **Physical activity** | | | | |
| Low level | 1.53 (0.79, 2.97) | 0.00 | 2.13 (0.98, 4.64) | 0.041* |
| High level | 1 ref | | | |

footwear worn most often, and physical activity for main effects, type of footwear by foot pain and BMI by foot pain for interaction effects.

Multivariate analysis identified that several independent variables were significantly associated with flatfoot; younger school children aged 11 and 12 years were about 3 times more likely (AOR: 3.27, 95% CI: 1.60, 6.65 and 3.30, 95% CI: 1.65, 6.62 respectively) to have a flatter foot, and 13 years old children were nearly two-folds likely (AOR: 1.97, 95% CI: 1.01, 3.97) to have flatfoot compared to 14–15 years age group. Male children were 1.5 folds more likely (AOR: 1.59, 95% CI: 1.03, 2.43) to have flatter foot than their counterparts. School children who self-reported to have suffered foot pain in the past 6 months were 2 folds more likely to have flat foot than those who reported no episodes of foot pain in the past 6 months. Compared to underweight the other BMI categories were very likely to have flatter foot; normal weight (AOR: 2.34, 95% CI: 1.21, 7.44), overweight (AOR: 3.7, 95% CI: 1.23, 8.71), obese (AOR: 4.16, 95% CI: 2.51, 10.96), and wearing closed shoes (AOR: 2.59, 95% CI: 1.57, 4.29) were significantly associated with flat foot.

The interaction effect between age-gender adjusted BMI category overweight, obese, and foot pain was significantly associated (overweight; AOR: 2.85, 95% CI: 1.62, 5.94, obese: 4.11, 95% CI: 2.85, 8.31). Wearing closed shoes by foot pain also showed significant interaction effects on flat foot (AOR: 2.59, 95% CI: 1.57, 4.29, p < 0.001). Given the vast difference between school types on the proportion of BMI category, subgroup regression analysis was considered. Although the subgroups (overweight, obese, by foot pain) remained significant in both subgroup models, the group sizes in certain categories were very small for adequate analysis Table 3.

## Discussion

This is the first study that examined the burden of flatfoot by using an objective measure in primary-school children aged 11–15 years and described the associated factors in Ethiopia, though a regional study reported on the adult population [24]. Presence of flatfoot was assessed using Staheli foot arch index using ink footprint method, which is simple, cost-effective, non-radiographic, can be safely used for larger samples in resource-limited setup, and recommended screening tool for flatfoot, could be an effective and reliable diagnostic tool as radiologic evaluation [25, 26].

The overall prevalence of flatfoot was 17.6% among children aged 11–15 years, which also means that more than one in every six children aged between 11–15 years has flatfoot, adding to the body of evidence that flatfoot is not an uncommon condition; there is need for an approach to prevent likely adulthood foot deformities. Interestingly, the prevalence of flatfoot was 17.5% among those near about 70% of normal-weight children included in this study. We also found that the prevalence of symptomatic flatfoot was 7.2%, flatfoot, flatfoot in most cases occurred bilaterally and that flexible flatfoot was more common. The variables associated with flatfoot or medial plantar arch development were age, sex, BMI, foot pain, type of footwear, and level of physical activity. The estimated prevalence of flatfoot of children aged 11–15 in our study is almost consistent with studies conducted in Iran (17.1, 16.1%), Colombia (15.7%), Islamabad, Pakistan (14.8%), and Sri Lanka (16.06%) [27–31]. It is hard to explain the reason for the similarity of estimations since these studies used different outcome methods and younger samples with different stature, but factors like the proportion of normal BMI, footwear's, and physical activity is similar to the study population. In contrast, the reported prevalence is lower than the findings that have been reported in higher socio-economic regions, Saudi Arabia (29.5%), Taiwan (59%), Poland (36%), Vienna, Austria (44%) and Nigeria (27.4%) [32–36]. In this study, larger proportions of children were normal weight and the mean weight of

participants was also lower compared to children living in developed countries. Another possible explanation could be due to the younger age of their participants the probability of a fat pad would have led to an overestimation of the prevalence. The estimation of the prevalence of flatfoot is influenced by many internal factors; age, gender, nutritional status, genetics, race, differential developmental milestones, and other external factors like; type of footwear, environmental conditions, and physical activity. Thereby resulting in disparate findings (0.6–77.9%) and a larger sample data from different areas of the world is needed [37]. In this study, the prevalence of flatfoot decreased with advancing age concord to results reported elsewhere [18, 31, 35, 38]. In particular, the prevalence plunged between 13 and 15 years. This might be attributed to the resolution or development of the medial arch with age. However, this hypothesis needs to be proven by a prospective cohort design. Our findings demonstrated that male children were marginally at higher risk of developing flatfoot than female children. This is consistent with the findings reported in the studies conducted in Taiwan, Greece, Nigeria, and Austria [33, 35, 39, 40], which could be partly explained by the greater rearfoot valgus and retarded development of rear foot in males compared with females. In contrast, a Nigerian study [41] found females to have a higher incidence of flatfoot in much older age, which might be due to a possible switch in gender preponderance between children and the adult population.

It is interesting to note underweight school children had lower odds of flatfoot in this study, compared to normal-weight children. But a previous report found underweight preschool children were at twofold at risk of having flatfoot. The divergent findings propose the importance of the factors of the developmental stage. In particular, foot pain among school children is associated with higher chances of flatfoot in this study. It is noteworthy that foot pain could result from strain resulting from excessive loading of midfoot or plantar pressure or deformed arch. Therefore, foot pain is a key clinical explanatory variable to understand predictive factors of flatfoot and for the development of early prevention strategies. The interaction between BMI and foot pain suggests that school children who are heavier and self-reported foot pain have a higher chance of developing flat foot. Furthermore, children with higher BMI were more prone to have symptomatic or painful flatfoot and activity-related foot pain. Similar findings were also reported by a study conducted in Iran [42]. Contrary to the findings of the Iranian study, in this study, symptomatic flatfoot was more prevalent among the younger age group and female children. Children wearing closed shoes were at higher risk of developing flatfoot than those wearing more exposed ones in this study. Proper choice and use of footwear early in life are directly relevant to foot health and important influencing factor in developing foot arches in children [43]. Moreover, tight-fitting, and rigid footwear are reported to provoke foot strain and deformities [44]. Supporting the findings of this study, a follow longitudinal study concluded that wearing closed shoes affects the longitudinal foot arch than those whose feet were more exposed [45]. The association of low level of physical activity and flatfoot in this study could be probably due to the condition of flat feet and/or foot pain might limit the desire for participation in locomotive activities that could aid weight management, somatic development, and healthy lifestyle. Not surprisingly, most of the overweight and obese children with symptomatic flatfoot in our study reported lower physical activity, which is a pivotal part of children's life. Subtalar arthroereisis has been reported as a minimal invasive and effective procedure in correction of flexible flatfoot, tarsal coalition, and accessory navicular syndrome mainly in children with flatfoot but also in adults [46]. But the availability of these procedures are scare in LMICs. Further, it is essential to understand that the measurement modalities used to diagnose flatfoot could influence the findings. Several footprint methods to calculate arch index like Cavanagh & Rodgers and Denis [47, 48] were also reported to be consistent along with the Staheli method. Notably, the weight-bearing radiographs have been

deemed to overcome the possible flaws of the footprint-based methods. Some authors even used a combination of these foot print methods for a more robust evaluation of flatfoot [42].

The findings of this study should be interpreted within the context of a few limitations. First, the implications suit the age range of our sample. Secondly, though the outcome measure used in this study better than the observational methods, it is also possible that the prevalence of flatfoot among overweight and obese children may not necessarily be flat, but fatter with more ground contact during footprint. However, exploratory radiological measures could be the subject of future studies to shed more light on children flatfoot. Thirdly, the cross-sectional nature did not account for the variance of recorded data in a typical week or month. In the absence of objective verification of pain among children, the potential for social desirability bias by children self-reporting pain may have occurred. Nevertheless, this study is a preliminary attempt with a relatively large sample in this country to report a well-powered insight and estimate the prevalence of flatfoot and related factors among Children. More importantly, the present study included about 80% of children who were underweight and normal weight to better explain the associations in the absence of imaging outcome measures.

## Conclusion

The foot posture in the developing children is certainly age-dependent and foot ink-print index measure is weight dependent. Considering the debate and growing knowledge in literature about fatter or flatter foot among fat and younger children while defining flatfoot based on ink foot-print index, beyond the limitation of foot-print index the prevalence of flatfoot among the large proportion of normal-weight children in this study are almost the same, which is a bothersome finding. There is certainly no evidence that osseous misaligned foot structure will auto-correct; in fact, they only progressively worsen if not intervened. Hence, symptoms of lower extremity pain, footwear use and choice, and relevant clinical examination shall be used to build a screening algorithm for diagnosis and treatment needs in children.

## Supporting information

**S1 File. Structured questionnaire in English version.**
(DOCX)

**S2 File. STROBE (STrengthening the Reporting of OBservational studies in Epidemiology) checklist for flatfoot among school children a school based cross-sectional study.**
(DOCX)

**S3 File. Correlation table showing the association between continuous variables and Staheli plantar arch index value.**
(DOCX)

## Acknowledgments

Firstly, we would like to express our deepest gratitude to the University of Gondar for fully funding this work. Our gratitude and appreciation goes to the data collectors, school children, teachers, and data collectors.

## Author Contributions

**Conceptualization:** Yohannes Abich, Balamurugan Janakiraman.

**Data curation:** Yohannes Abich, Balamurugan Janakiraman.

**Formal analysis:** Yohannes Abich, Moges Gashaw, Balamurugan Janakiraman.

**Investigation:** Yohannes Abich, Tewodros Mihiret.

**Methodology:** Yohannes Abich, Tewodros Mihiret, Temesgen Yihunie Akalu.

**Project administration:** Yohannes Abich, Balamurugan Janakiraman.

**Resources:** Tewodros Mihiret, Temesgen Yihunie Akalu, Moges Gashaw.

**Software:** Temesgen Yihunie Akalu.

**Supervision:** Temesgen Yihunie Akalu, Balamurugan Janakiraman.

**Validation:** Moges Gashaw.

**Writing – original draft:** Tewodros Mihiret, Moges Gashaw, Balamurugan Janakiraman.

**Writing – review & editing:** Temesgen Yihunie Akalu, Balamurugan Janakiraman.

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
