## [Decision Letter · Decision Letter 0]

15 Jul 2020

PONE-D-20-14474

Flatfoot and associated factors among Ethiopian school children aged 11 to 15 years: a school-based study

PLOS ONE

Dear Dr. Janakiraman,

Thank you for submitting your manuscript to PLOS ONE. After careful consideration, we feel that it has merit but does not fully meet PLOS ONE’s publication criteria as it currently stands. Therefore, we invite you to submit a revised version of the manuscript that addresses the points raised during the review process.

Please address the points raised by reviewer 2 and the presentation issues identified by both reviewers.

We look forward to receiving your revised manuscript.

Kind regards,

Alison Rushton

Academic Editor

PLOS ONE

"This study was fully funded by the University of Gondar (Grant no: SOM 1251/2019). The

views presented in the article are the authors and not necessarily express the views of the funding

organization. University of Gondar did not involve in the design of the study, data collection,

analysis, and interpretation."

4. Your ethics statement must appear in the Methods section of your manuscript. If your ethics statement is written in any section besides the Methods, please move it to the Methods section and delete it from any other section. Please also ensure that your ethics statement is included in your manuscript, as the ethics section of your online submission will not be published alongside your manuscript.

Reviewers' comments:

Reviewer's Responses to Questions

**Comments to the Author**

1. Is the manuscript technically sound, and do the data support the conclusions?

Reviewer #1: Yes

Reviewer #2: Yes

2. Has the statistical analysis been performed appropriately and rigorously? 

Reviewer #1: Yes

Reviewer #2: Yes

3. Have the authors made all data underlying the findings in their manuscript fully available?

Reviewer #1: Yes

Reviewer #2: No

4. Is the manuscript presented in an intelligible fashion and written in standard English?

Reviewer #1: Yes

Reviewer #2: Yes

5. Review Comments to the Author

Reviewer #1: 

This manuscript was well written and the data analysis was more than adequate. I fully agree that more attention needs to be given to the pathology of pediatric flatfeet and that they should not be considered "normal." There were a few spacing errors and missing "." but that would be caught later in the editing process. I feel this paper is ready to proceed to the next step.

Reviewer #2: 

Thank you for allowing me to revise this paper dealing with flatfoot in Ethiopian school children. This study involved over 800 children and found a prevalence of 17.6% for flatfoot. Also, overweight, obesity and pain in children were correlated with a greater chance of having flatfoot.

The introduction is clear and nicely written. Methods are overall sound and the statistical analysis is appropriate. Results are well presented as well. In discussion there are multiple typos which need to be addressed. Apart from this, concepts are clear. My only remark would be to mention further methods to define flatfoot, which is still a matter of debate, such as the Denis classification and the Cavanagh one which are mentioned in the paper "The role of arthroereisis of the subtalar joint for flatfoot in children and adults." EFORT Open reviews 2017, highlighting the differences and the advantages of one method as compared to one other.

6. PLOS authors have the option to publish the peer review history of their article (what does this mean?). If published, this will include your full peer review and any attached files.

Reviewer #1: **Yes: **Michael E. Graham, DPM

Reviewer #2: No

---

## [Decision Letter · Decision Letter 1]

7 Aug 2020

Flatfoot and associated factors among Ethiopian school children aged 11 to 15 years: a school-based study

PONE-D-20-14474R1

Dear Dr. Janakiraman,

We’re pleased to inform you that your manuscript has been judged scientifically suitable for publication and will be formally accepted for publication once it meets all outstanding technical requirements.

Kind regards,

Alison Rushton

Academic Editor

PLOS ONE

Reviewers' comments:

Reviewer's Responses to Questions

**Comments to the Author**

1. If the authors have adequately addressed your comments raised in a previous round of review and you feel that this manuscript is now acceptable for publication, you may indicate that here to bypass the “Comments to the Author” section, enter your conflict of interest statement in the “Confidential to Editor” section, and submit your "Accept" recommendation.

Reviewer #2: All comments have been addressed

2. Is the manuscript technically sound, and do the data support the conclusions?

Reviewer #2: Yes

3. Has the statistical analysis been performed appropriately and rigorously? 

Reviewer #2: Yes

4. Have the authors made all data underlying the findings in their manuscript fully available?

Reviewer #2: Yes

5. Is the manuscript presented in an intelligible fashion and written in standard English?

Reviewer #2: Yes

6. Review Comments to the Author

Reviewer #2: The paper has been nicely amended, I have no further remarks. I think the paper is ready for publication.

7. PLOS authors have the option to publish the peer review history of their article (what does this mean?). If published, this will include your full peer review and any attached files.

Reviewer #2: **Yes: **Alessio Bernasconi, MD PhD FEBOT

---

## [Editor Report · Acceptance letter]

13 Aug 2020

PONE-D-20-14474R1 

Flatfoot and associated factors among Ethiopian school children aged 11 to 15 years: a school-based study 

Dear Dr. Janakiraman:

I'm pleased to inform you that your manuscript has been deemed suitable for publication in PLOS ONE. Congratulations! Your manuscript is now with our production department. 

Kind regards, 

on behalf of

Professor Alison Rushton 

Academic Editor

PLOS ONE